https://doi.org/10.1038/s41467-020-15017-1　　OPEN

# A riboswitch gives rise to multi-generational phenotypic heterogeneity in an auxotrophic bacterium

Jhonatan A. Hernandez-Valdes [1], Jordi van Gestel [2,3,4,5] & Oscar P. Kuipers [1]✉

Auxotrophy, the inability to produce an organic compound essential for growth, is widespread among bacteria. Auxotrophic bacteria rely on transporters to acquire these compounds from their environment. Here, we study the expression of both low- and high-affinity transporters of the costly amino acid methionine in an auxotrophic lactic acid bacterium, *Lactococcus lactis*. We show that the high-affinity transporter (Met-transporter) is heterogeneously expressed at low methionine concentrations, resulting in two isogenic subpopulations that sequester methionine in different ways: one subpopulation primarily relies on the high-affinity transporter (high expression of the Met-transporter) and the other subpopulation primarily relies on the low-affinity transporter (low expression of the Met-transporter). The phenotypic heterogeneity is remarkably stable, inherited for tens of generations, and apparent at the colony level. This heterogeneity results from a T-box riboswitch in the promoter region of the *met* operon encoding the high-affinity Met-transporter. We hypothesize that T-box ribo- switches, which are commonly found in the Lactobacillales, may play as-yet unexplored roles in the predominantly auxotrophic lifestyle of these bacteria.

[1] Department of Molecular Genetics, Groningen Biomolecular Sciences and Biotechnology Institute, University of Groningen, 9747 AG Groningen, Netherlands. [2] Department of Evolutionary Biology and Environmental Studies, University of Zürich, Zürich, Switzerland. [3] Swiss Institute of Bioinformatics, Lausanne, Switzerland. [4] Department of Environmental Microbiology, Swiss Federal Institute of Aquatic Science and Technology (Eawag), Dübendorf, Switzerland. [5] Department of Environmental Systems Science, ETH Zürich, Zürich, Switzerland. ✉email: o.p.kuipers@rug.nl

Many bacteria in nature are auxotrophic: they lack functional biosynthetic pathways to synthesize organic compounds that are essential for growth[1]. Amino acid auxotrophy is among the most common form of auxotrophies[2,3]. Bacteria that lost the capacity to synthesize certain essential amino acids depend on their environment (e.g., other bacterial species or their eukaryotic host) to obtain the missing compounds. Auxotrophy is expected to evolve when amino acids are abundant in the environment and can readily be taken up. Biosynthetic pathways can either be lost through selection against the futile costs of expressing these pathways or through the accumulation of neutral mutations that gradually deteriorate them[4,5]. Since auxotrophies can give rise to ecological dependencies, where one bacterium relies on another for growth, they are proposed to strongly shape the interactions between cells inside bacterial communities[6]. Indeed, multiple studies have shown how syntrophic cross-feeding interactions, based on the reciprocal exchange of amino acids, lead to stable coexistence[2,7]. Auxotrophies are therefore likely to play a prominent role in determining the composition and stability of microbial communities[8–10].

Auxotrophic bacteria can obtain the essential amino acids by either directly sequestering freely available amino acids from their environment or through the enzymatic breakdown of environmental proteins. For example, in mixed-culture dairy fermentations of *Lactococcus lactis* strains, bacteria initially compete for free amino acids available in milk (isoleucine, leucine, valine, histidine and methionine are essential for most of the *L. lactis* strains), and subsequently compete for peptides by releasing proteases that break down the casein molecules[11,12]. Fluctuations in the availability of external amino acids caused by competition between auxotrophic bacteria or otherwise, are expected to favor optimized uptake strategies to sequester environmental amino acids as efficiently as possible, in order to assure that the auxotrophic bacterium can continue to grow. Different kinds of membrane transporters are important for amino acid uptake, ranging from generic low-affinity transporters that facilitate the uptake of several different amino acids (i.e., broad substrate specificity) to targeted high-affinity transporters that can import specific amino acids only at high efficiency[13,14].

Here, we study how an auxotrophic bacterium regulates amino acid uptake in response to different levels of amino acid availability in the environment. Specifically, we focus on the uptake of methionine in *L. lactis*, a well-studied lactic acid bacterium. Since methionine is one of the most costly amino acids to synthesize, methionine auxotrophies are commonly found in nature[15]. *L. lactis* presumably lost the capacity to synthesize methionine de novo during adaptation to milk[16]. *L. lactis* can use two different transporters to import methionine: a high-affinity ABC transporter (named in this study Met-transporter) and a low-affinity transporter named the branched-chain amino acid permease (BcaP). The low-affinity transporter primarily transports branched-chain amino acids (BCAA: isoleucine, leucine, valine), but can also transport methionine and to a lesser extent cysteine[17]. We start our analysis by studying the expression of the high-affinity transporter under a range of methionine concentrations. When methionine is limiting, we observe strong heterogeneity in the expression of the Met-transporter: whereas some cells show high expression of the Met-transporter, others express the same transporter only weakly. Cells with weak expression rely on the low-affinity BcaP-transporter to acquire enough methionine. Interestingly, the differential expression of the Met-transporter is stably inherited across tens of generations, due to which heterogeneity is also apparent at the colony level. We analyze thousands of these colonies to quantify the heterogeneous gene expression at different methionine concentrations and subsequently study the regulatory underpinnings that give rise to this heterogeneity. We demonstrate that a T-box riboswitch plays a critical role in the emergence of the phenotypic heterogeneity in methionine uptake.

## Results

**Phenotypic heterogeneity at the single-cell level**. We start by analyzing the expression of the high-affinity transporter of methionine, i.e., the ABC-transporter of the methionine uptake transporter family in *L. lactis*. The genome of *L. lactis* MG1363 encodes a single Met-transporter in the *met* operon, which resembles the *metNPQ* operon of *Bacillus subtilis*[18], but is composed of six genes: four genes encode homologous ATP-binding proteins (*plpA, plpB, plpC,* and *plpD*), one encodes a permease (*ydcB*) and one encodes a lipoprotein (*ydcC*). To visualize the expression of the Met-transporter at different methionine concentrations, we fused the *met* promoter (Pmet) to a gene encoding for a green fluorescent protein (*gfp*). The resulting strain, *L. lactis* Pmet-gfp, shows that the Met-transporter is expressed at the population level in a concentration dependent way: at lower methionine concentrations the expression levels of the *met* operon are higher (Fig. 1a). Next, we examine *L. lactis* Pmet-gfp cells using time-lapse fluorescence microscopy in the standard chemically defined medium (CDM) that contains 0.27 mM methionine[19]. To our surprise, cells show a strongly heterogeneous expression of the *met* operon, whereas some cells exhibit a high expression of the *met* operon (GFP+ cells), others show a low expression (GFP− cells) (Fig. 1b). Heterogeneity could also be observed at lowest possible methionine concentration that supports stable growth in CDM (0.025 mM), but was absent at higher (1 mM) methionine concentrations (Fig. 1c; see also Supplementary Fig. 1). In other words, the subpopulation of GFP+ cells, with high *met* expression, disappears at high methionine concentrations (Fig. 1d), corresponding to the low expression levels observed at the population level (Fig. 1a). Interestingly, the time-lapse experiment also shows that both GFP− and GFP+ cells continue to proliferate over time without changing fluorescence levels, which suggests that phenotypic heterogeneity is stably inherited across generations (Fig. 1b and Supplementary Movies 1–3).

**Stable phenotypic heterogeneity across colonies**. Since the time-lapse microscopy shows stable phenotypic inheritance, we next studied the expression of the *met* operon at the colony level. To this end, we grew colonies on agar plates of CDM supplemented with a range of different methionine concentrations (0.025–10 mM) (Fig. 2, Supplementary Fig. 2) and analyzed the expression of the *met* operon. In total, we quantified expression levels in more than 8000 colonies using automatic image analysis (see Methods and Supplementary Figs. 2, 3). In agreement with the single-cell data (Fig. 1), also at the colony level, we observe clear heterogeneity in the expression of the Met-transporter at low methionine concentrations (Fig. 2a, c). This finding confirms that phenotypic heterogeneity is indeed stably inherited across numerous of generations, as shown in the time-lapse experiment (Fig. 1b). At high methionine concentrations, the heterogeneity in *met* expression disappears (Fig. 2b, c). In order to quantify the phenotypic heterogeneity in more detail, we categorize the colonies based on their expression level (see Methods): GFP+ colonies with high *met* expression and GFP− colonies with low *met* expression (Fig. 2d). Figure 2e shows that the fraction of GFP+ colonies across the different methionine concentrations follows a step function, having ~45% GFP+ colonies at the lowest methionine concentrations and only GFP− phenotype at the highest concentrations.

We also analyzed *met* expression of cells within individual GFP+ and GFP− colonies at both low (0.025 mM) and high (10 mM)

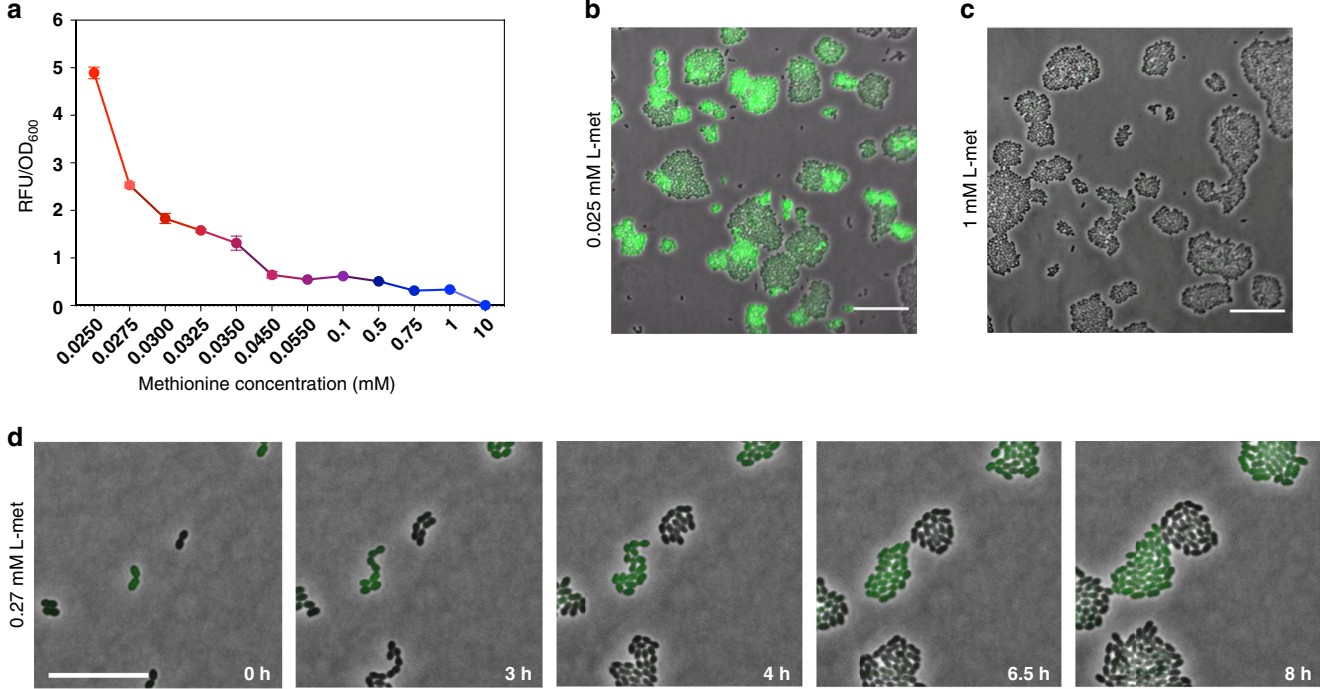

**Fig. 1 The Met-transporter is heterogeneously expressed at low methionine concentrations in *L. lactis*. a** Expression of the Met-transporter at the population level in cultures of the *L. lactis* Pmet-gfp strain, grown in CDM supplemented with increasing concentrations of methionine (0.025 mM to 10 mM). Data are presented as mean ± S.D. Error bars represent standard deviation (SD) of the mean values of three independent experiments. Source data are provided as a Source Data file. **b** Two phenotypes coexist when *L. lactis* Pmet-gfp is grown in standard CDM containing methionine at concentration of 0.27 mM. Snapshots of a time-lapse fluorescence microscopy experiment, where the GFP+ cells (green cells) reflect high *met* expression and the GFP− cells (black cells) show low *met* expression. Scale bar, 15 μm. **c, d** Snapshots of single-cell fluorescence microscopy, when the cells are grown in CDM with low and high methionine concentrations (0.025 and 1 mM) respectively (Supplementary Movies 2 and 3). Overlays of green-fluorescence and phase-contrast images are shown. Scale bars, 15 μm.

methionine concentrations to determine if within-colony expression levels are homogeneous. Specifically, we collected individual colonies, resuspended them in phosphate-buffered saline solution, and used flow cytometry to quantify the fluorescence of the constituent cells. Figure 2f shows that no heterogeneity in fluorescence levels could be detected within colonies, meaning that all cells homogeneously show either high *met* expression (GFP+ colonies) or low *met* expression (GFP− colonies). Cells in GFP− colonies grown at 10 mM methionine show lower fluorescence levels than cells from GFP− colonies at 0.025 mM. This observation shows that the latter colonies weakly express the Met-transporter. The lack of phenotypic heterogeneity inside colonies confirms that cells rarely switch phenotype and are committed to either a high or low expression of the Met-transporter for numerous generations. Only sometimes, we observed phenotypic switches, which are apparent at the colony level through sector formation (e.g., inset of Fig. 2g and Supplementary Fig. 4). At the lowest methionine concentration, less than 3% of the colonies show signs of switching and this fraction declines significantly with higher methionine concentrations (Fig. 2g). Despite being so extremely rare, we did manage to capture a few phenotypic switches between low and high expression of the *met* operon at the single-cell level using time-lapse microscopy (Supplementary Movie 4).

To make sure that the rare phenotypic switches are not caused by genetic mutations, we sequenced the upstream region of the *met* operon in both GFP+ and GFP− sectors in switching colonies, as well as in homogeneous GFP+ and GFP− colonies. No mutations were detected in the *met* promoter region. To also rule out mutations elsewhere, we also performed whole-genome sequencing on the same colony samples (GFP+, GFP−, and

sectored colonies; see Supplementary Fig. 5) and also there no mutations were observed. These results confirm our notion that the heterogeneous expression of the *met* operon and rare phenotypic switches between GFP+ and GFP− cells have a phenotypic origin.

Given the stability of the phenotypic heterogeneity, we were curious to see if *met* expression had any effect on colony growth by analyzing the sizes of both GFP+ and GFP− colonies. Figure 3a shows that, at low methionine concentrations, the GFP+ colonies are significantly larger than GFP− colonies, although the difference is minimal. This result suggests that high expression of the Met-transporter provides an advantage when methionine concentration are low, but also shows that high *met* expression is not strictly necessary to support growth on low methionine concentrations. At high methionine concentrations the advantage of high *met* expression disappears, as indicated by the large colony sizes of GFP− colonies (GFP+ colonies are absent at this concentration). Additionally, we also quantified the GFP+ and GFP− colonies using flow cytometry, which allows us to assess generation times. We calculated the number of generations of each colony based on the total number of cells present after 48 h of colony growth (Methods; Supplementary Table 3), assuming that a single cell founded each colony. The flow cytometry data also suggests that cells inside GFP+ colonies grow slightly quicker than cells in GFP− colonies at low methionine concentrations (Fig. 3b), although in this case the difference is not significant. Taken together, these results are consistent with the hypothesis that the high-affinity transporter facilitates growth when methionine is limited. Moreover, the low fraction of colonies with phenotypic switches (Fig. 2g) in combination with the number of generations within each colony (Fig. 3b), suggests that

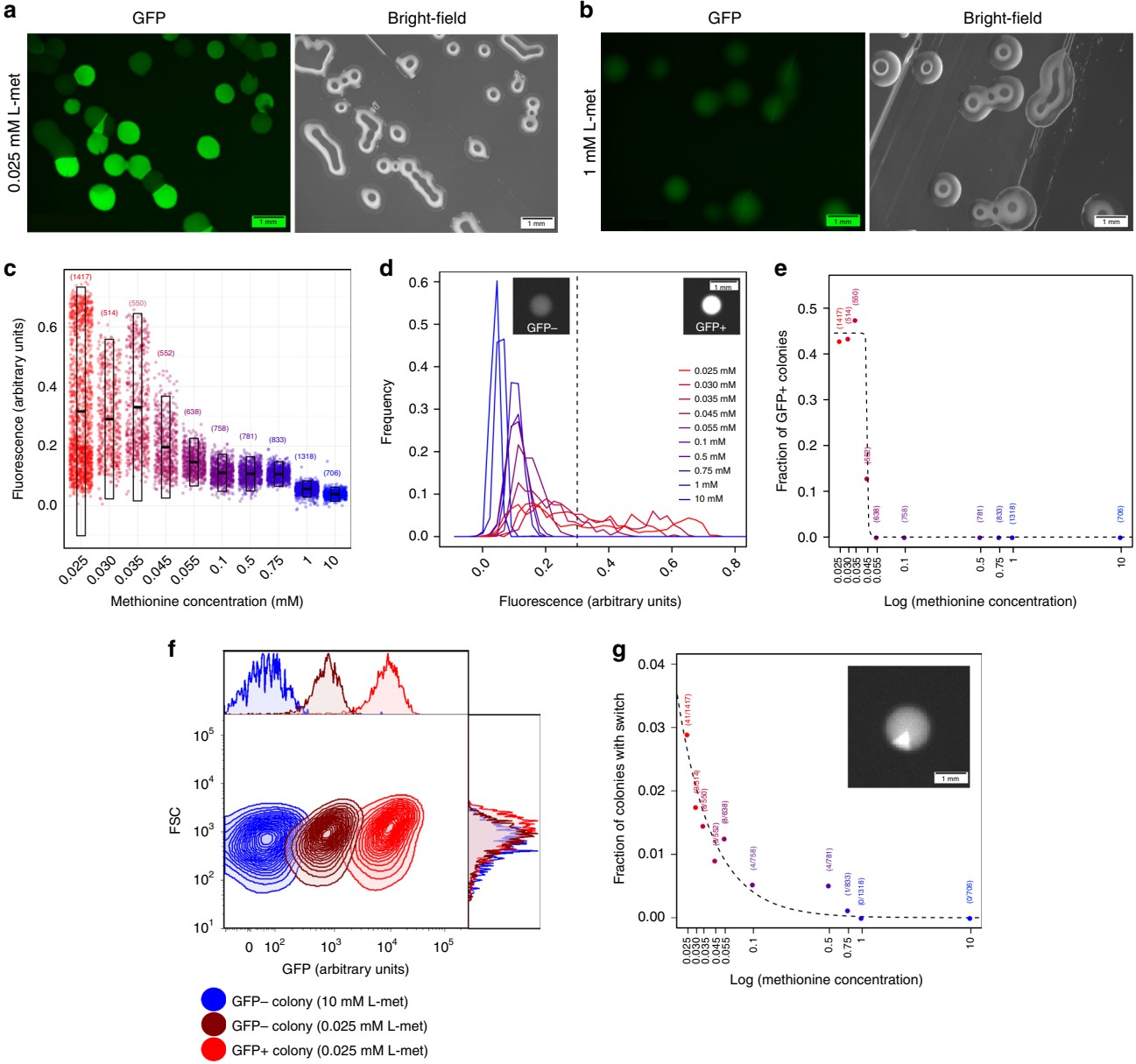

**Fig. 2 The phenotypic heterogeneity at the colony level.** *L. lactis* colonies grown for 48 h on CDM-agar plates with either, **a**, low (0.025 mM) or, **b** high (1 mM) methionine concentrations. **a**, **b** left image shows green-fluorescence channel and right image shows bright-field channel. Scale bars, 1 mm. **c** Mean fluorescence intensities of individual colonies grown on CDM-agar plates at different methionine concentrations (0.025–0 mM, red to blue; see also Supplementary Figs. 2, 3). Transparent boxes show mean and standard deviation, and number between parentheses shows number of analyzed colonies. **d** Distribution of mean fluorescence intensities across colonies for different methionine concentrations (0.025–10 mM, red to blue). Vertical dotted line demarcates colonies categorized as having low *met* expression (GFP−) or high *met* expression (GFP+). Scale bar, 1 mm, **e** Fraction of GFP+ colonies at different methionine concentrations. Dotted line shows fit of step function ($y = a\text{-}a/(1 + e^{-100\cdot(x+b)})$; $a = 0.450 \pm 0.007$ (s.e.), $P_a < 10^{-10}$, $b = 3.110 \pm 0.001$, $P_b < 10^{-16}$). **f** Fluorescence measurements by flow cytometry show the *met* expression in each type of colony phenotype: GFP+ colony (light-red) grown on low methionine concentrations (0.025 mM), GFP− colony (dark-red) grown on low methionine concentrations (0.025 mM), and GFP− colony (blue) grown on high methionine concentrations (10 mM). 10,000 ungated events for each sample are shown. **g** Switching rate at different methionine concentrations (0.025 mM to 10 mM, red to blue). Inset shows example of colony with switch in expression level. Numbers in parenthesis show number of colonies with expression switch and total number of analyzed colonies. Dotted line shows exponential fit ($y = e^{-a\cdot(x-b)}$; $a = 1.34 \pm 0.27$ (s.e.), $P_a < 0.01$; $b = 6.41 \pm 0.61$, $P_b < 10^{-5}$). Scale bar, 1 mm. Source data are provided as a Source Data file.

phenotypic heterogeneity is stably inherited for (at least) tens of generations.

**The role of transporters in the origin of heterogeneity.** We expect that the GFP− cells, which only weakly express the Met-transporter at low methionine concentrations, rely on the branched-chain amino acid permease (BcaP) to acquire enough methionine. We therefore hypothesize that in the absence of this low-affinity transporter, all cells homogeneously express the high-affinity Met-transporter when methionine is limiting. Figure 4 shows the expression of the *met* operon in the absence of the low-affinity transporter (Δ*bcaP*) at both low and high methionine concentrations (see also Supplementary Fig. 6). Indeed, the

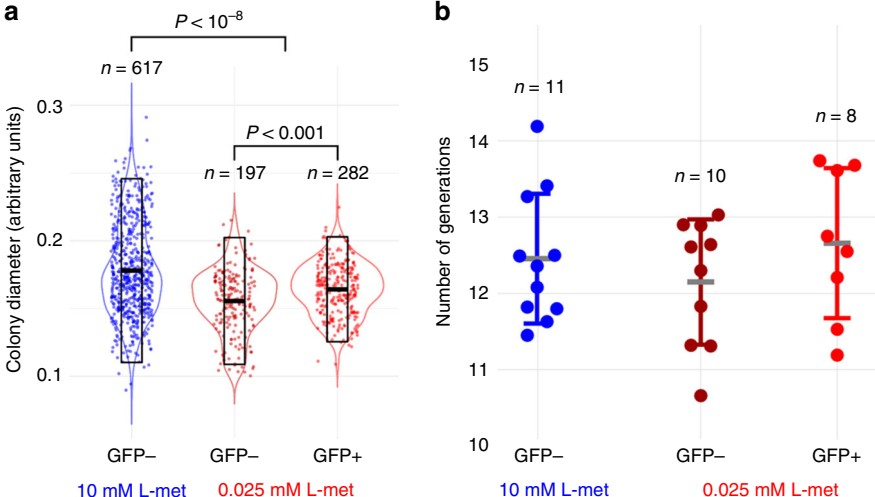

**Fig. 3 The phenotypic states are inherited in long-term. a** Diameter of colonies on CDM-agar plates with high (10 mM) and low (0.025 mM) methionine concentrations. At high methionine concentrations, only GFP− colonies are observed (blue). At low methionine concentrations, both GFP− (dark red) and GFP+ (red) colonies are observed. Transparent boxes show mean and standard deviation. $P < 10^{-8}$ (10 mM GFP− vs. 0.025 GFP−/GFP+), $P < 10^{-3}$ (0.025 mM GFP− vs. 0.025 mM GFP+). Statistically significant based on two-tailed Mann–Whitney $U$ test. **b** Number of generations in colonies as determined by flow cytometry. Although the number of generations inferred from the flow cytometry data in **b** follow the same trend as the colony diameter data in (**a**), differences were not significant (two-tailed Mann–Whitney test). Data are presented as mean ± S.D. Error bars represent standard deviation (SD).

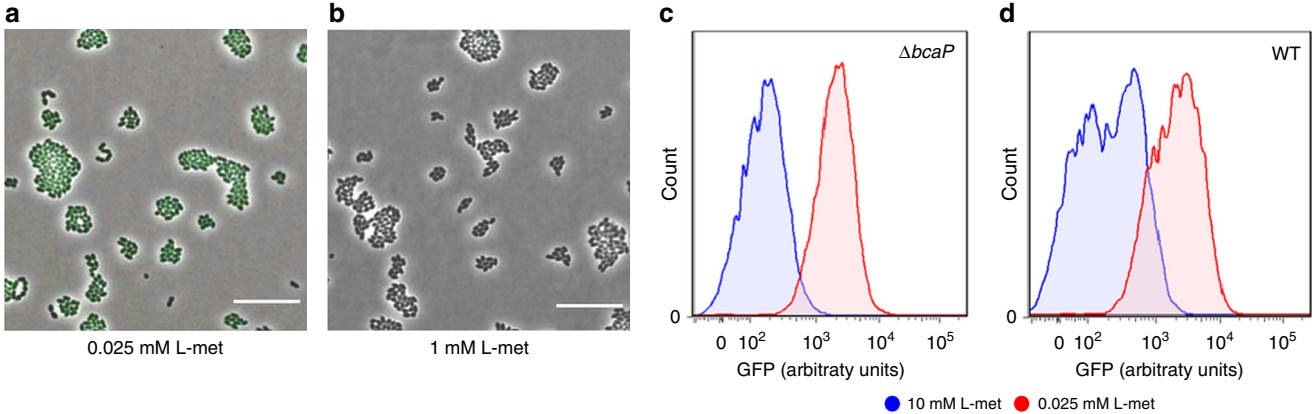

**Fig. 4 The low-affinity methionine transporter, BcaP, supports the growth of the GFP− cells. a, b** Snapshots of single-cell fluorescence microscopy in the *L. lactis ΔbcaP Pmet-gfp* strain. **a** cells grown at low methionine concentrations (0.025 mM) and **b** at high methionine concentrations (1 mM). Scale bars, 15 μm. **c, d** Single-cell fluorescence measurements by flow cytometry in the *bcaP* deletion mutant strain (**c**; *L. lactis ΔbcaP Pmet-gfp*) and wild-type strain (**d**; *L. lactis Pmet-gfp*), grown in CMD with low and high methionine concentrations (0.025 and 1 mM, in red and blue respectively), 10,000 ungated events recorded are shown. Source data are provided as a Source Data file.

deletion of *bcaP* resulted in the homogeneous expression of the high-affinity Met-transporter (Fig. 4a). In accordance, the flow cytometry data shows a clear separation in fluorescence levels between populations grown at low and high methionine concentrations for the *L. lactis ΔbcaP* strain, but not for the wild type, where expression is heterogeneous (Fig. 4c, d). These results support our expectation that the GFP− cells in the wild-type strain utilize the low-affinity transporter to compensate for the relative low expression of the Met-transporter. This compensation likely results from increased expression levels of the low-affinity transporter. To test this, we constructed a *codY* knockout mutant. CodY is a transcriptional repressor of *bcaP*[20]. Its knockdown therefore increases the expression of the low-affinity transporter. As expected, the *codY* knockout mutant shows a decreased *met* expression (Supplementary Fig. 11), thereby confirming that the low-affinity transporter can directly compensate for the high-affinity transporter.

In agreement with the homogeneous expression of the *met* operon in *L. lactis ΔbcaP* strain, the deletion of the *met* operon results in a homogeneous expression as well (Supplementary Fig. 7, Supplementary Movies 5, 6). Importantly, the *L. lactis Δmet* strain requires higher concentrations of methionine for growth, confirming that the Met-transporter is essential for growth on low methionine concentrations. This finding corroborates our results in Fig. 3, which shows that GFP+ colonies have a slight growth advantage over GFP− colonies at low methionine concentrations (see also Supplementary Figs. 7–9). In the absence of a high-affinity transporter, all cells become dependent on the low-affinity BcaP-transporter for methionine uptake. At this transporter, methionine encounters competition with the other branched-chain amino acids, which are transported via BcaP as well[17]. Consequently, higher methionine concentrations are acquired to sustain growth. The competition between methionine and the branched-chain amino acids is also

apparent in the wild type, where *bcaP* is over-expressed when *L. lactis PbcaP-gfp* is grown at high methionine concentrations, but normalizes again when the other branched-chain amino acids are provided at high concentrations as well (Supplementary Fig. 10).

**Global regulators and phenotypic heterogeneity.** To further disentangle how the phenotypic heterogeneity in *met* expression is brought about, we next focus on the role of global regulators. It has been shown before that heterogeneity can result from cells that enter distinct physiological states[21,22]. In general, Gram-positive bacteria employ two global regulators to regulate amino acid uptake: CodY and Rel[23,24]. CodY is a transcription factor that represses the expression of amino acid transporters when amino acids are abundant (like the effect of CodY on *bcaP* expression mentioned above)[25,26]. Rel is a bifunctional protein that can both synthesize and degrade phosphorylated purine-derived alarmones (p)ppGpp[27]. In this way, Rel can activate the so-called stringent response, which is a general stress response triggered by nutrient stress[28], such as amino acid starvation that is sensed through uncharged tRNAs. Besides CodY and Rel, also the carbon catabolite repression, regulated by CcpA, has been linked to amino acid uptake through its indirect effect on sulfur metabolism[29,30]. We investigated whether CodY, CcpA and Rel also affect the expression of the *met* operon by deleting either *codY, ccpA* or *rel* from the chromosome of *L. lactis*. Interestingly, the gene deletions strongly reduce *met* expression at low methionine concentrations (Supplementary Fig. 11), showing that the *met* operon is affected by global expression changes during amino acid starvation. Moreover, in all cases we see a loss of the bimodal distribution in *met* expression, and hence a loss of GFP− and GFP+ cells. The most striking results were obtained for the *rel* mutant, which reduced *met* expression most strongly and gave rise to expression distributions that exactly match those of the GFP− cells in the wild type (across various methionine concentrations; Supplementary Fig. 11). This finding suggests that the stringent response is essential for the appearance of the GFP+ subpopulation. Since Rel does not directly bind to the *met* promoter, the effect of Rel on *met* expression is probably indirect. Therefore we next examine transcriptional regulation of the *met* operon specifically.

**Transcriptional regulation of the *met* operon.** In the closely related streptococci, genes underlying the uptake and metabolism of sulfur amino acids (methionine and cysteine) are known to be controlled by three LysR- family regulators, MetR/MtaR, CmbR and HomR[31]. The MetR/MtaR regulator activates a *met*-like operon, where promoter binding is triggered by homocysteine (L-HC)[32,33]. The genome of *L. lactis* MG1363 encodes one homologue of MetR, called CmhR. Similar to MetR in the streptococci, CmhR is predicted to have a binding site in the promoter region of the *met* operon[34]. To examine if CmhR indeed affects *met* expression, we determined expression of the *met* operon in a *cmhR* knockout strain. In contrast to the deletion of the global regulators, which only reduce expression, the deletion of CmhR completely precludes expression of the *met* operon. This indicates that, similar to its homolog in streptococci[33,35], CmhR directly regulates *met* expression (Fig. 5a, Supplementary Fig. 13). Since L-HC was shown to facilitate promoter binding[35], we also evaluated the effect of L-HC on the *met* expression. Figure 5b shows that under low methionine concentrations (0.025 mM), the addition of L-HC strongly increases *met* expression (Supplementary Fig. 14). Yet, surprisingly, the addition of L-HC left the phenotypic heterogeneity practically unchanged, i.e., both populations with low and high fluorescence levels are still observed

(Fig. 5c), which shows that CmhR activity is not responsible for the observed heterogeneity.

In addition to transcription factors, many methionine transporters in the Bacilli and Clostridia are regulated by riboswitches[18,36]. We therefore examined if a riboswitch could mediate the observed heterogeneity. Based on sequence analysis (see Methods), we identified a single regulatory element (RE) in the leader region of the *met* operon that, with high confidence, corresponds to a T-box riboswitch (Supplementary Fig. 15). This is a well-known regulatory element that occurs in many lactobacilli and staphylococci[37–39] and has been detected in other *L. lactis* strains before[34]. The T-box monitors amino acid availability by discriminating between charged and uncharged tRNA, and effectively up-regulates gene expression when the amino acid associated with the sensed tRNA is limiting (that is, when uncharged tRNAs are abundant)[40]. Interestingly, in our case, the T-box riboswitch is predicted to specifically respond to uncharged tRNA$^{Met}$ based on sequence similarity to other T-box elements[34]. We therefore hypothesize that under low methionine availability, the presence of uncharged tRNA$^{Met}$ could facilitate the expression of the *met* operon, thereby giving rise to high expression levels and potentially phenotypic heterogeneity.

Figure 5d, e shows that the deletion of the RE sequence of the *met* promoter results in homogeneous Met-transporter expression (see also Supplementary Fig. 17a). The lack of the RE sequence delimitates the regulation of the *met* promoter to the activity of the CmhR transcription factor. At low methionine concentrations, CmhR promotes the expression of the *met* operon. Conversely, at high concentrations, expression of the *met* operon decreases, although some expression remains, probably because of residual CmhR activity that is triggered by the cellular pool of homocysteine (Supplementary Fig. 16). In contrast, in the wild type where the RE is present, low methionine concentrations result in a fraction of cells that strongly express the *met* operon (GFP+ cells), whereas the remaining cells have the same expression level (GFP− cells) as the mutant strain that lacks the RE. On the contrary, at high methionine concentrations, the expression of the *met* operon is more strongly suppressed in the wild type than in the mutant strain that lacks the RE. These results are consistent with the regulatory architecture of a T-box riboswitch, where high levels of uncharged tRNA$^{Met}$ stimulate the expression of the *met* operon at low methionine concentrations, and low levels of uncharged tRNA$^{Met}$ at high concentrations prevent its expression.

To further delineate the role of the T-box riboswitch in the origin of phenotypic heterogeneity, we next examine highly targeted mutations inside the conserved domains of the riboswitch. Fig. 6a visualizes the various domains in the T-box riboswitch at the 5' UTR of the *met* operon, based on homology with previously studied riboswitches[41–43] (see Supplementary Fig. 17). In total, we examine four targeted mutations: Mutant 1 (G72T, G73T) targets the stability of the stem I domain, which is known to interact with the tRNA in order to ensure binding and recognition of the cognate tRNA ligand[41,44]; Mutant 2 (ΔUGGU) is a deletion of the T-box sequence, which are the nucleotides where the uncharged tRNA binds and triggers the formation of the anti-terminator complex[45]; Mutant 3 (C258U, G259U, U260C) targets conserved nucleotides of the anti-terminator conformation[43], thereby affecting its stability; Mutant 4 is a 60 bp deletion (Δ306-365) in the terminator sequence, preventing the formation of the terminator hairpin.

We compare *met* expression in each of the four mutants to that of the wild type (*L. lactis Pmet-gfp*) across a range of methionine concentrations (0.025, 0.035, 0.5, 1, 10 mM). In contrast to the wild type, all riboswitch mutants show a single expression peak and, thus, the absence of phenotypic heterogeneity. Mutation 1

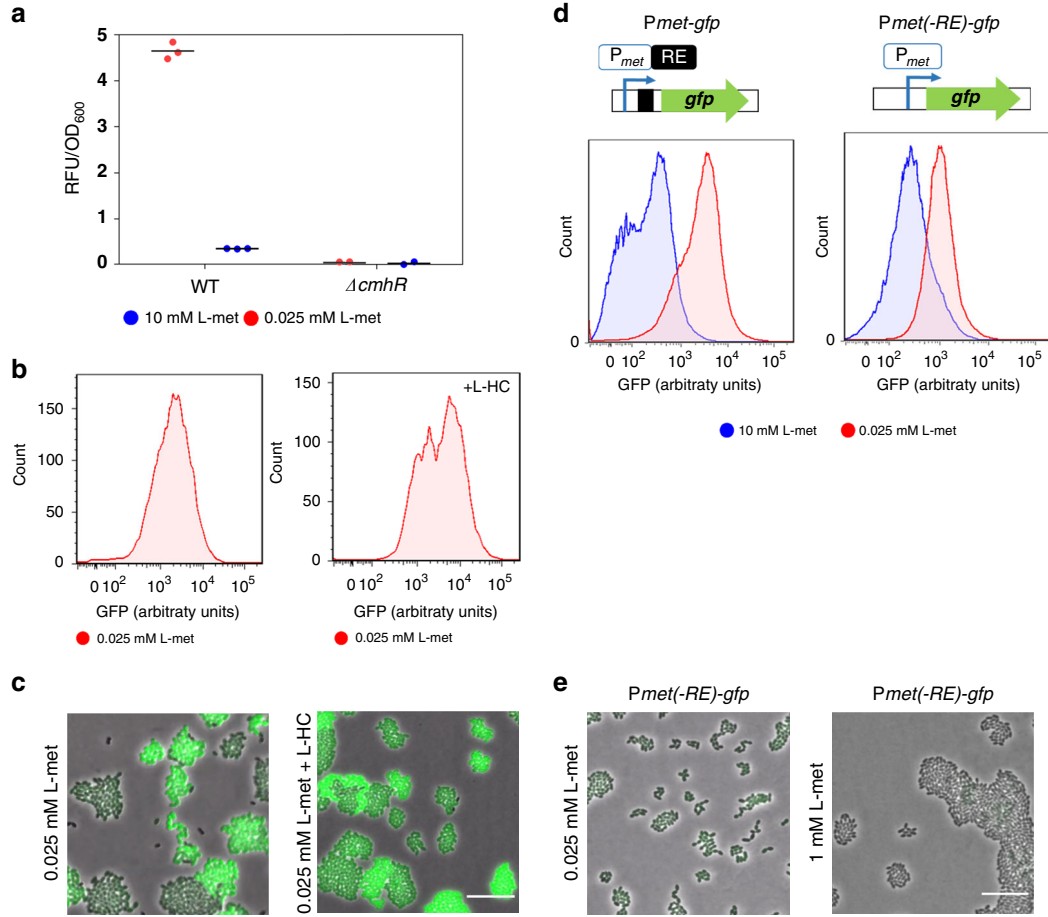

**Fig. 5 Transcriptional regulation of the *met* operon. a** Expression of the Met-transporter at the population-level, in the *cmhR* deletion mutant (*L. lactis ΔcmhR Pmet-gfp*) and wild-type (*L. lactis Pmet-gfp*) strains at low (0.025 mM) and high (1 mM) methionine concentrations. Data are presented as mean ± S.D. Error bars represent standard deviation (SD) of the mean values of three independent experiments. **b** Single-cell fluorescence measurements by flow cytometry, in the *L. lactis Pmet-gfp* strain in the absence (left) and presence of 0.27 mM L-homocysteine (L-HC; right) (in both cases with 0.025 mM methionine; see Supplementary Movies 7 and 8). 10,000 ungated events for each sample are shown. **c** Snapshots of single-cell fluorescence microscopy in the *L. lactis Pmet-gfp* strain in the presence (left) and absence of 0.27 mM L-homocysteine (L-HC; right). Scale bar, 15 μm. **d** Single-cell fluorescence measurements by flow cytometry, in the *met* promoter-driven expression of GFP with (*L. lactis Pmet-gfp*; left) or without the regulatory element in the *met* promoter (*L. lactis Pmet(-RE)-gfp*; right) at low (0.025 mM) and high (1 mM) methionine concentrations. 10,000 ungated events for each sample are shown. **e** Snapshots of single-cell fluorescence microscopy in the *L. lactis Pmet(-RE)-gf*p strain at low (0.025 mM; left), and high (1 mM; right) methionine concentrations. Scale bar, 15 μm. Source data are provided as a Source Data file.

and 3 show the weakest *met* expression. In these mutations the riboswitch is entirely dysfunctional, which results in transcriptional attenuation by the terminator hairpin. Interestingly, mutant 2 and 4 give expression distributions that closely match that of the GFP− and GFP+ subpopulations in the wild type, respectively (at low methionine concentrations). These results are in agreement with the T-box tRNA sensing mechanism[43]. Namely, in the absent of key residues in the T-box (Mutant 2), uncharged tRNA cannot bind the riboswitch, which prevents the formation of the anti-terminator complex. As a consequence, transcriptional attenuation by the terminator hairpin lowers *met* expression. Conversely, in the absence of the terminator hairpin (Mutant 4), transcriptional attenuation does not occur. As a consequence, *met* expression solely depends on the activity of the transcription factor (CmhR), which results in high *met* expression at low methionine concentrations (0.025 mM) and low *met* expression at high methionine concentrations (10 mM) (see also Supplementary Fig. 18). Altogether, these results confirm that the T-box riboswitch is directly responsible for the observed phenotypic heterogeneity in the wild type, giving rise to GFP− and GFP+ subpopulations.

We finally also tested if the effect of the stringent response on *met* expression, as studied by the *rel* knockout above (Supplementary Fig. 11), exerts its effect through the riboswitch (as opposed to the transcription factor, CmhR) by examining a double knockout mutant (Supplementary Fig. 12). This revealed that *rel* deletion mutant only results in the loss of the GFP+ subpopulation in the presence of the riboswitch (i.e., regulatory element), showing that the stringent response affects *met* expression via the riboswitch only.

In summary, our results show that the *met* operon is subject to three hierarchical layers of regulation (see Fig. 7); at the highest level, (i) global regulators indirectly affect *met* expression, by changing the expression of large suites of genes involved in amino acid uptake and biosynthesis; (ii) then, at a lower level, CmhR affects the expression of the *met* operon by specifically binding the promoter region (the same regulator is also involved in the uptake and metabolism of other sulfur amino acids); and, at the lowest level, (iii) a T-box riboswitch regulates expression of the *met* operon, by responding to methionine starvation, thereby giving rise to a remarkably stable form of phenotypic heterogeneity. Together, these regulatory layers orchestrate how the

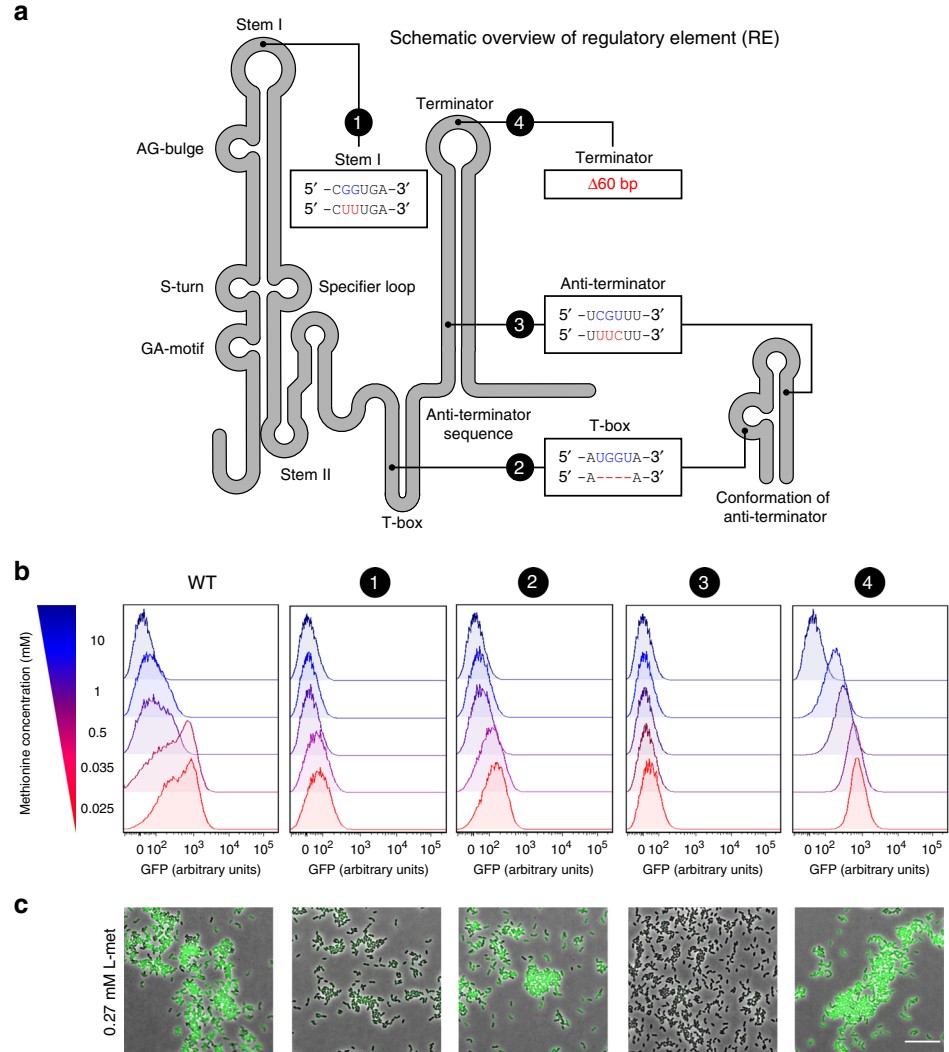

**Fig. 6 Regulation of the methionine sensing by structural elements of the T-box riboswitch. a** Secondary structure diagram of the regulatory element (RE; T-box riboswitch) used in this study. The four introduced mutations to the T-box are shown in red and with numbers (1–4). Highly conserved domains in T-box riboswitches are indicated. **b** Single-cell fluorescence measurements by flow cytometry, in the *met* promoter-driven expression of GFP in the *L. lactis Pmet-gfp* (WT) and each of the four mutants of the regulatory element in the *met* promoter (1–4), grown in CDM supplemented with increasing concentrations of methionine (0.025–10 mM; red to blue). 10,000 ungated events for each sample are shown. Source data are provided as a Source Data file. **c** Snapshots of single-cell fluorescence microscopy in all the *L. lactis Pmet-gfp* strains, grown in standard CDM (0.27 mM methionine). Scale bar, 15 µm.

auxotrophic *L. lactis* cells respond to methionine starvation; enabling cells to sequester sufficient amounts of methionine when concentrations are low.

## Discussion

Auxotrophic bacteria often rely on transporters to acquire essential organic compounds from their environment. Here, we studied the regulation of methionine uptake by examining the expression of low- and high-affinity transporters in *L. lactis*, a well-studied lactic acid bacterium that is auxotrophic for this amino acid. We reveal an extraordinary case of long-term phenotypic heterogeneity. Under methionine-limited conditions, *L. lactis* differentiates into two phenotypic subpopulations (Fig. 7): one subpopulation imports free methionine from the environment by a high-affinity transporter (GFP+), whereas the other one is primarily sustained by the uptake via a low-affinity transporter (GFP−). These phenotypes are remarkably stable and inherited for tens of generations.

Our data support the following origin of phenotypic heterogeneity (Fig. 7): Upon methionine depletion, cells need to increase methionine uptake rates to sustain growth. In some cells, the repression of *bcaP* by CodY[20] is released timely, resulting in the upregulation of the low-affinity transporter, which allows cells to sequester enough methionine from the environment to support growth. These cells also weakly express the high-affinity transporter, through transcriptional activation of CmhR. As a consequence, this first subpopulation (GFP−) of timely responding cells will not experience methionine starvation (Fig. 7a). In contrast, cells that do not respond timely will experience methionine starvation, which leads to the accumulation of uncharged tRNA$^{Met}$ (Fig. 7b). These tRNAs bind to the riboswitch[41], where they trigger formation of the anti-terminator complex that prevents transcriptional attenuation by the terminator hairpin[40,46]. In addition, tRNAs trigger the stringent response[47,48], leading to a physiological state in which cells presumably stay locked[21,49,50] for several generations. Ultimately, the increased expression of the high-affinity transporter increases the rate of methionine uptake and, thereby, supports growth in the second subpopulation of cells (GFP+).

**Fig. 7 Proposed model of methionine uptake and phenotypic heterogeneity in *L. lactis*.** Free methionine can enter to the cell by two ways, via a low-affinity BcaP-transporter (branched-chain amino acid permease) and through a high-affinity Met-transporter (composed of the PlpABCD, YdcC and YdcB proteins). The *met* operon is regulated by CmhR, using homocysteine (L-HC) as coeffector. The expression of *bcaP* is controlled by CodY, which is activated by branched-chain amino acids (BCAA). At low methionine concentrations, the cells increase methionine uptake rates to sustain growth and two colony phenotypes are observed: GFP− colonies (**a**; left) and GFP+ colonies (**b**; right). **a** In GFP− colonies the repression of *bcaP* by CodY is released timely, and the upregulation of the low-affinity transporter results in enough methionine to support growth. Although GFP− colonies weakly express the Met transporter as well. **b** In GFP+ colonies, methionine starvation leads to the presence of uncharged tRNA$^{Met}$, and this signal strongly up-regulates the expression of the *met* operon via the regulatory element (RE; T-box riboswitch) in the leader region of the *met* mRNA, which assures transcriptional continuation. In addition, the uncharged tRNA$^{Met}$ triggers the stringent response due to amino acid starvation.

Over the last decades, riboswitches have been shown to play a central role in amino acid biosynthesis and uptake[51–54]. To our knowledge, this study is the first to show that regulation at the RNA level, namely the T-box at the 5′UTR of the *met* operon, can give rise to phenotypic heterogeneity. Comparative genomic studies have shown that T-box regulation strongly expanded in the Lactobacillaceae family (i.e., lactic acid bacteria), where it replaced the S-box riboswitch (Supplementary Table 4) that is triggered by S-adenosyl-L-methionine[36,55]. This same phylogenetic group is also characterized by extensive gene loss, presumably because the ancestor underwent a lifestyle switch from being mostly free-living to a host-associated lifestyle[56,57]. Loss was particularly common among genes underlying biosynthetic pathways; not surprisingly, many lactic acid bacteria are therefore auxotrophic for methionine synthesis, including *Lactococcus lactis*[9,58], *Lactobacillus plantarum*[9], *Lactobacillus helveticus*[59], *Streptococcus pyogenes*[60], *Streptococcus thermophilus*[61], and *Enterococcus faecalis*[62]. In accordance with this lifestyle switch, the lactic acid bacteria are also characterized by an exceptional broad repertoire of carbon and amino acid transporters[56], indicating that these bacteria often live in nutrient-rich environments. We postulate that the T-box regulation underlying some of these transporters might have evolved to accommodate the largely auxotrophic lifestyle of the lactic acid bacteria, where the T-box riboswitch might provide a more immediate regulatory response to amino acid starvation than the S-box riboswitch due to its specificity for methionine or any of the other auxotrophic amino acids[63,64].

Why is methionine uptake heterogeneously expressed? Although phenotypic heterogeneity could simply arise as a side-product of regulation that is effectively neutral or even deleterious, it can also convey a few important advantages. First,

phenotypic heterogeneity could support a bet-hedging strategy[65]. When the distinct phenotypes in a heterogeneous population provide benefits under different environmental conditions, a population could prepare itself for unexpected changes in the environment by expressing both phenotypes: e.g., cells that strongly express the Met-transporter might do better in environments with limited methionine, whereas cells that weakly express the Met-transporter might do better when the amino acid becomes more abundant. Bet-hedging has been reported in *L. lactis* before, where different cell types that appear during the diauxic shift are prepared to consume alternative future carbon sources[21]. Second, phenotypic heterogeneity could also support a division of labor[66,67], where cells that either strongly or weakly express the Met-transporter engage in a cooperative interaction that benefits them both. At this stage, we can only speculate what such cooperative benefits might be, but one can imagine that cells might not only differ in methionine uptake, but also in a number of other metabolic traits[21]. If so, this could allow for some form of metabolic division of labor, where subpopulations exchange metabolites in the same way as synthetic bacterial communities can exchange amino acids[8,68–72]. Exploring these and other potential benefits of the long-term phenotypic heterogeneity in methionine uptake is an exciting topic for future research.

## Methods

**Bacterial strains and growth conditions.** We used the *Lactococcus lactis* MG1363 (ref. [73]) strain in this study. *L. lactis* cells were grown at 30 °C in M17 broth (Difco$^{TM}$ BD, NJ, USA) or in CDM[19], supplemented with glucose (Sigma-Aldrich) 0.5% (w/v). CDM contained 49.6 mM NaCl, 20.1 mM Na$_2$HPO$_4$, 20.2 mM KH$_2$PO$_4$, 9.7 μM (±)-α-lipoic acid, 2.10 μM D-pantothenic acid, 8.12 μM nicotinic acid, 0.41 μM biotin, 4.91 μM pyridoxal hydrochloride, 4.86 μM pyridoxine hydrochloride, 2.96 μM thiamine hydrochloride, 0.24 μM (NH$_4$)$_6$Mo$_7$O$_{24}$, 1.07 μM CoSO$_4$, 1.20 μM CuSO$_4$, 1.04 μM ZnSO$_4$, 20.12 μM FeCl$_3$, 1.46 mM L-alanine, 1.40 mM L-arginine, 0.61 mM

L-asparagine, 1.03 mM L-aspartic acid, 0.35 mM L-cysteine, 0.66 mM L-glutamic acid, 0.66 mM L-glutamine, 0.39 mM glycine, 0.16 mM L-histidine, 0.63 mM L-isoleucine, 0.89 mM L-leucine, 1.02 mM L-lysine, 0.27 mM L-methionine, 0.39 mM L-phenylalanine, 3.58 mM L-proline, 1.64 mM L-serine, 0.57 mM L-threonine, 0.18 mM L-tryptophan, 2.76 mM L-tyrosine and 0.73 mM L-valine.GM17-agar or CDM-agar plates were prepared by adding agar 1.5% (w/v) and glucose to M17 or CDM, respectively. When necessary, culture media was supplemented with erythromycin (Sigma-Aldrich, MO, USA) 5 μg mL$^{-1}$.

*E. coli* DH5α (Life Technologies, Gaithersburg, MD, USA) was used to perform all the recombinant DNA techniques. Cells were grown at 37 °C in Luria-Bertani broth or Luria-Bertani agar 1.5% (w/v) (Difco™ BD, NJ, USA). For screening of colonies containing recombinant plasmids, erythromycin 150 μg mL$^{-1}$ was added.

For microscopy experiments and plate-reader assays, *L. lactis* cells were grown in CDM with glucose 0.5% (w/v) and collected by centrifugation from exponential growth cultures (optical density of 0.3 at 600 nm) and washed three times with phosphate-buffered saline (PBS) solution (pH 7.2) containing: 15.44 μM KH$_2$PO$_4$, 1.55 mM NaCl and 27.09 μM Na$_2$HPO$_4$.

### Recombinant DNA techniques and oligonucleotides.

DNA amplifications by PCR were performed using a PCR mix containing Phusion HF Buffer (Thermo Fisher Scientific Inc., MA, USA), 2.5 mM dNTPs mix, Phusion HF DNA polymerase (Thermo Fisher Scientific Inc., MA, USA), primers (0.5 μM each), and 50 ng of *L. lactis* chromosomal DNA as template. Oligonucleotides (Supplementary Table 1) were purchased from Biolegio (Nijmegen, The Netherlands). PCRs were performed in an Eppendorf thermal cycler (Eppendorf, Hamburg, Germany). The DNA target sequence of interest was amplified by 35 cycles of denaturation (98 °C for 30 s), annealing (5 °C or more lower than $T_m$ for 30 s), and extension (70 °C for 1 min per 1 Kbp). Amplifications were confirmed by 1 % agarose gel electrophoresis method.

For DNA cloning, we used Fast-digest restriction enzymes and T4 DNA ligase (Thermo Fisher Scientific Inc., MA, USA). Reactions were performed according to the manufacturer's recommendations. The ligation products were transformed into *E. coli* DH5α (Life Technologies, Gaithersburg, MD, USA) competent cells by electroporation. Cells were plated on Luria-Bertani agar plates with appropriate antibiotics and grown overnight at 37 °C. Screening of colonies to confirm the genetic construct was performed by colony PCR. Positive colonies with correct constructs were inoculated in Luria-Bertani broth with the appropriate antibiotic. Plasmid DNA and PCR products were isolated and cleaned-up with a high pure plasmid isolation kit (Roche Applied Science, Mannheim, Germany), according to the protocol of the manufacturer. DNA sequences of constructs were always confirmed by DNA sequencing (Macrogen Europe, Amsterdam, The Netherlands).

### Construction of the *L. lactis* gfp strains.

All constructed strains are described in Supplementary Table 2. To construct the vector pSEUDO::Pmet-gfp, carrying the *L. lactis* MG1363 *met* promoter, the promoter region was amplified by PCR using the oligonucleotides metFw and metRv, using chromosomal DNA as template. The PCR fragment was cleaved with *Pae*I/*Xho*I enzymes and ligated to pSEUDO-gfp[74]. The vector pSEUDO::Pmet-gfp was introduced in *L. lactis* MG1363 via electroporation[75]. The vector was integrated into the silent *llmg_pseudo10* locus of *L. lactis* by a single-crossover integration. Transformants were selected on M17-agar plates supplemented with sucrose, glucose and erythromycin 5 μg mL$^{-1}$, yielding the *L. lactis* Pmet-gfp strain. The vector pSEUDO::Pmet-gfp was introduced by electroporation in *L. lactis* MG1363 Δ*met*, *L. lactis* MG1363 Δ*codY*[20], *L. lactis* MG1363 Δ*rel* (a kind gift of Saulius Kulakauskas), *L. lactis* MG1363 Δ*cmhR* (a kind gift of Anne de Jong), *L. lactis* MG1363 Δ*ccpA*[30], *L. lactis* MG1363 Δ*bcaP*[25] and *L. lactis* MG1363 Δ*bcaP*Δ*brnQ*[25].

To construct the plasmid pSEUDO::Pmet(-RE)-gfp, carrying the *L. lactis* MG1363 *met* promoter, but lacking the regulatory element (RE) of the promoter region (the T-box riboswitch was completely deleted), the *met(-RE)* promoter was amplified by PCR using the oligonucleotides metFw and met(-RE)_Rv, using chromosomal DNA as template. The PCR fragment was cleaved with *Pae*I/*Xho*I enzymes and ligated to pSEUDO-gfp. Chromosomal integration in *L. lactis* MG1363 at the *llmg_pseudo10* locus was performed as described above, and the *L. lactis* Pmet(-RE)-gfp strain was obtained. The same vector pSEUDO::Pmet(-RE)-gfp was introduced by electroporation in *L. lactis* MG1363 Δ*rel*.

To construct the plasmid pSEUDO::PbcaP-gfp, carrying the *L. lactis* MG1363 *bcaP* promoter, the promoter region was amplified by PCR using the oligonucleotides bcaP_Fw and bcaP_Rv, using chromosomal DNA as template. The PCR fragment was cleaved with *Pae*I/*Xho*I enzymes and ligated to pSEUDO-gfp[74]. The vector pSEUDO::PbcaP-gfp was integrated into the *llmg_pseudo10* locus of *L. lactis* MG1363 by single-crossover recombination. Transformants were selected on M17-agar plates supplemented with sucrose, glucose and erythromycin 5 ug mL$^{-1}$, yielding the *L. lactis* PbcaP-gfp strain.

The T-box mutants: mutant 1 (G72T, G73T), mutant 2 (ΔUGGU), mutant 3 (C258U, G259U, U260C) and mutation 4 (Δ306-365), were constructed by using two strategies. The first strategy consists of site directed mutagenesis of whole plasmid[76]. PCRs were performed with mutagenic oligonucleotides carrying the desired mutation in form of mismatches to the original plasmid and using the the vector pSEUDO::Pmet-gfp as DNA template. The oligonucleotides containing the desired mutations: Mut1-Fw and Mut1-Rv (mutant 1), Mut2-Fw and Mut2-Rv

(mutant 2), Mut3-Fw and Mut3-Rv (mutant 3) are listed in Supplementary Table 1. The vectors containing the desired mutations: *Pmet(mut1)-gfp*, *Pmet(mut2)-gfp*, and *Pmet(mut3)-gfp*, were obtained in *E. coli* and confirmed by DNA sequencing. After confirmation, the vectors were integrated into the *L. lactis* genome as described above, yielding the Mutant 1, Mutant 2 and Mutant 3 strains. The second strategy was used to obtain Mutant 4 strain. This mutant lacking 60 bp of the terminator was obtained by PCR amplification using the oligonucleotides metFw and Mut4-Rv using chromosomal DNA as template. The PCR fragment was cleaved with *Pae*I/*Xho*I enzymes and ligated to pSEUDO-gfp. Chromosomal integration in *L. lactis* MG1363 at the *llmg_pseudo10* locus was performed as described above, and the Mutant 4 strain was obtained.

### Gene manipulation.

Gene deletion mutants were obtained by homologous recombination using the system based on homologous recombination with pCS1966 (ref. [77]). To delete the native promoter of the *met* operon, upstream and downstream regions of the promoter region were amplified using the oligonucleotides: A_metKO_Fw, A_metKO_Rv, B_metKO_Fw, and B_metKO_Rv. The fragment A obtained (PCR product using the oligonucleotides A_metKO_Fw and A_metKO_Rv) was ligated into pCS1966 via *Kpn*I/*Eco*RI restriction sites. The plasmid obtained was named pCS1966-A. Fragment B (PCR product using the primers B_metKO_Fw and B_metKO_Rv) was cloned into pCS1966-A via *Bam*HI/*Not*I restriction sites, and the plasmids obtained was named pCS1966-AB. All recombinant pCS1966 were initially constructed in *E. coli* DH5a (Life Technologies) and then introduced to *L. lactis*. The *L. lactis* MG1363 strain was transformed with the vector pCS1966-AB via electroporation. Homologous recombination in two-steps was performed by growing *L. lactis* cells in SA medium plates[78] supplemented with 30 ug mL$^{-1}$ 5-fluoroorotic acid hydrate (Sigma-Aldrich). The deletion mutant strain *L. lactis* Δ*met* was confirmed by PCR and sequencing of a PCR fragment in the genomic region of interest (Macrogen Europe, Amsterdam, The Netherlands).

### RNA extraction, RT-PCR, and quantitative RT-PCR.

Total RNA was isolated from *L. lactis* MG1363 wild type and *bcaP* deletion mutant, grown overnight in CDM as described above. *L. lactis* cells were diluted 1:20 in CDM supplemented with different methionine concentrations (0.025 and 10 mM). Cells were harvested at late exponential phase at an optical density at 600 nm (OD$_{600}$) of 0.35. RNA was isolated with the High Pure RNA isolation kit (Roche Life Science; Penzberg, Germany). Assessment of RNA integrity and purity was performed by running an aliquot of the RNA sample on a denaturing agarose gel stained with ethidium bromide (EtBr), and by quantification using a NanoDrop™ Spectrophotometer. The RNA samples were treated with 2 U of DNase I (Invitrogen, UK).

For qRT-PCR, reverse transcription of the RNA samples was performed with the SuperScript™ III Reverse Transcriptase kit (Thermo Fisher Scientific Inc., MA, USA). Quantitative PCR analysis was performed using an iQ5 Real-Time PCR Detection System[79] (Bio-Rad Laboratories, CA, USA). Reactions were performed using a master mix containing SsoAdvanced universal SYBR® Green supermix 2 × (Bio-Rad), 2.5 mM dNTPs mix (Thermo Scientific), primers (0.5 μM each), and 100 ng of cDNA as template. Oligonucleotides (Supplementary Table 1) were purchased from Biolegio (Nijmegen, The Netherlands).The transcription level of the *met* operon was normalized to *rpoE* and *rarA* transcription. The amplification was performed with oligonucleotides: Fw-qRT_PCR, Rv2-qRT-pCR, and Rv8-QRT-pCR (*met* operon); rarA-Fw and rarA-Rv (*rarA* gene), and rpoE_Fw and rpoE-Rv (*rpoE* gene). The results were interpreted using the comparative C$_T$ method[80].

### Time-lapse microscopy experiments.

Washed cells were transferred to a solidified thin layer of CDM with high-resolution agarose 1.5% (w/v) (Sigma-Aldrich, MO, USA). When appropriate, CDM without methionine was used, and supplemented with varying concentrations of methionine (L-methionine; Sigma-Aldrich, MO, USA) or L-homocysteine (Sigma-Aldrich, MO, USA). A standard microscope slide was prepared with a 65 μL Gene Frame AB-0577 (1.5 × 1.6 cm) (Thermo Fisher Scientific Inc., MA, USA). A 30 μL volume of heated CDM-agar was set in the middle of the frame and covered with another microscope slide to create a homogeneous surface after cooling. The upper microscope slide was removed and bacterial cells were spotted on the agar. The frame was sealed with a standard microscope coverslip.

Microscopy observations and time-lapse recordings were performed with a temperature-controlled (Cube and box incubation system Life Imaging Services) DeltaVision (Applied Precision, Washington, USA) IX7I microscope (Olympus, PA, USA), at 30 °C. Images were obtained with a CoolSNAP HQ2 camera (Princeton Instruments, NJ, USA) at ×60 or ×100 magnification. 300-W xenon light source, bright –field objective and GFP filter set (filter from Chroma, excitation 470/40 nm and emission 525/50 nm). Snapshots in bright-field and GFP-channel were taken every 10 min for 20 h with 10% APLLC while LED light and a 0.05 s exposure for bright–filed, or 100% xenon light and 0.8 s of exposure for GFP-signal detection. The raw data was stored using softWoRx 3.6.0 (Applied precision) and analyzed using ImageJ software[81].

**Microscopy observations in bacterial colonies**. *L. lactis* cells were grown overnight in CDM, washed three times in PBS, streaked on CDM-agar plates containing varying concentrations of methionine and incubated at 30 °C for 48 h. The fluorescence in *L. lactis* colonies was detected using an Olympus MVX20 macro zoom fluorescence microscope equipped with a PreciseExcite light-emitting diode (LED) fluorescence illumination (470 nm), GFP filter set (excitation 460/480 nm and emission 495/540 nm). Images were acquired with an Olympus XM10 monochrome camera (Olympus Co., Tokyo, Japan).

**Colony analysis**. Microscopy images were analyzed in Matlab R2018a using automatic image analysis (see also Supplementary Figs. 2, 3). The algorithm consists of five steps: (1) Images are first converted to a black and white images to identify regions with potential colonies, using the *im2bw* function; (2) Wrongly segmented regions are automatically removed, based on size and shape; (3) Individual colonies are then detected by fitting circles to the segmented image regions, using the *imfindcircles* function; (4) All identified colonies are subsequently inspected, to manually remove all wrongly detected colonies and to annotate which colonies show signs of switching (see inset of Fig. 2g and Supplementary Fig. 4); (5) Finally, summary statistics of the manually curated set of colonies are collected, including colony size, colony location and average fluorescence intensity of colony. In this way, we acquired data for more than 8000 individual colonies. For Fig. 3b, we acquired all measurements of the colony diameters manually using Fiji 1.51d[82], to avoid potential biases that could be introduced by the algorithm by irregularly shaped colonies (importantly, however, qualitatively similar results are obtained based on automatic size detection).

**DNA sequencing**. The bacterial colonies of each phenotype were resuspended in 15 μL of PBS, and 1 μL of the colony suspension was used as template to perform colony PCR. The *met* promoter was amplified by colony PCR using the oligonucleotides met_promoter_Fw and met_promoter_Rv. The DNA sequences and the reference *met* promoter sequence were aligned with Clustal Omega[83].

The genomes of all different colonies were paired-end sequenced at the Beijing Genomics Institute (BGI, Copenhagen N, Denmark) on a BGISEQ-500 platform. A total of 5 million paired-end reads (150 bp) were generated. FastQC version 0.11.5 (ref. [84]) was used to examine the quality of the reads. Identification of mutations was performed with Breseq[85], using the *Lactococcus lactis* subsp. *cremoris* MG1363, complete genome, as a reference sequence (GenBank: AM406671.1).

**Plate-reader assays**. Cultures of *L. lactis* were grown and prepared as described above. For fluorescence intensity measurements, *L. lactis* cells were diluted 1:20 in CDM. When testing the effect of varying concentrations of methionine, CDM was used and supplemented with different methionine concentrations. The growth and fluorescence signal were recorded in 0.2 mL cultures in 96-well micro-titer plates and monitored by using a micro-titer plate reader VarioSkan (Thermo Fisher Scientific Inc., MA, USA). Growth was recorded with measurements of the optical density at 600 nm ($OD_{600}$) and the GFP-signal was recorded with excitation 485 nm and emission 535 nm every 10 min for 24 h. Both signals were corrected for the background value of the corresponding medium used for growth. The GFP-signals in relative fluorescence units (RFU) were normalized by the corresponding $OD_{600}$ measurements yielding $RFU/OD_{600}$ values.

**Flow cytometry**. *L. lactis* cultures were grown overnight in CDM as described above, washed three times in PBS and transferred to fresh CDM supplemented with varying concentrations of methionine. The cultures were incubated at 30 °C and samples were taken either at exponential or stationary growth phase. A threshold for the FSC and SCC parameters was set (200 in both) in the FACS Canto flow cytometer (BD Biosciences, CA, USA) to remove all the events that do not correspond to cells. The GFP-signal at all the measured cells was recorded in 10,000 events and used for downstream analysis (named ungated events in the corresponding figures). GFP-signal measurements were obtained with a FACS Canto flow cytometer (BD Biosciences, CA, USA) using a 488 nm argon laser. Raw data was collected using the FACSDiva Software 5.0.3 (BD Biosciences). And the FlowJo software was used for data analysis (https://www.flowjo.com/).

For the analysis of single colonies by flow cytometry, *L. lactis* were grown in CDM-agar plates with varying concentrations of methionine. After incubation at 30 °C for 48 h, the colonies are randomly chosen, and carefully taken out by using a pipette tip. We cut off the end of p200 tips to create an agar cutter and isolate single colonies. The isolated colonies were vigorously suspended in 400 μL PBS. A constant volume of 120 μL was analyzed by flow cytometry to calculate the number of cells in each colony and the GFP-signal at single-cell level. The generation number ($n$) was calculated with the formula $n = \log X / \log 2$ where $X$ is the total number of cells in the colony, considering that the initial number of cells in each colony is one (see Supplementary Table 3).

**Statistics and reproducibility**. Statistical analyses were performed using Prism 6.01 (GraphPad software https://www.graphpad.com/) and R v3.3.0. All experiments were repeated independently at least three times. All micrographs, including small insets, show representative images from three independent replicate experiments.

**Bioinformatics**. The regulatory elements (T-box riboswitch) in promoter regions were identified using RibEX[86] and RegPrecise 3.0 database[34]. Transcription factor binding-motifs identified in the *met* promoter region were analyzed with PePPER[87]. Alignments and sequences identities were determined by using Clustal 2.1 using the full-length protein or DNA sequences[88]. Identification of mutations was performed with breseq version 0.32.1.

**Reporting summary**. Further information on research design is available in the Nature Research Reporting Summary linked to this article.

## Data availability
Data supporting the findings of this work are available within the paper and its Supplementary Information files. The source data underlying Figs. 1a, 2c–e, 2f, 3a, 3b, 4c–d, 5a, 5c, 5d and 6b, and Supplementary Figs. 1, 5b, 6a, 6b, 7c, 7d, 8, 9, 10a, 11a, 11b, 12a, 12b, 13a, and 14a are provided as a Source Data file. All other data are available from the corresponding author on request.

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

## Acknowledgements

We thank Ana Solopova (PC Microbiome Institute, University College Cork) for helpful discussions. We thank Anne de Jong and Danny Incarnato (both Department of Molecular Genetics, University of Groningen) for their help with sequence analysis. J.A.H.V. and O.P.K. were financed by the Netherlands Organization for Scientific Research (NWO), research program TTW (13858). J.v.G. received support from the EMBO Long-Term Fellowship (ALTF 1101-2016) and the Marie Sklodowska-Curie Individual Fellowship (742235).

## Author contributions

J.A.H.V. and O.P.K. conceived the study. J.A.H.V. designed and carried out all the experiments. J.v.G. designed and performed the colony analysis. J.A.H.V. and J.v.G. analyzed the data. J.A.H.V, J.v.G., and O.P.K. wrote the manuscript. J.v.G. and O.P.K. provided supervision. All authors discussed the results and commented on the manuscript.

## Competing interests

The authors declare no competing interests.
