## [Peer Review File · Nature Communications]

Reviewers' comments:

Reviewer #1 (Remarks to the Author):

The authors describe phenotypic heterogeneity in bacterial population that has very low switching rate and therefore the phenotypes are maintained over several generations. Although cases of phenotypic heterogeneity have been described previously, the rather stable phenotypes described in the manuscript are novel.

Therefore the paper would be of interest for researchers of the specific field and also for wider audiences.

The manuscript is very well written. As the Results part is easy to follow I found the beginning of Discussion (lines 288-316) that describes the findings in the manuscript too long and repetitive. The authors can consider shortening this section.

An important detail concerns the RelA protein. The RelA was originally described in *Escherichia coli* and there it is not a bifunctional enzyme. Unfortunately some of the diverse bifunctional homologues of RelA that are present in diverse range of bacteria are sometimes also named RelA. This creates considerable confusion. The *Lactococcus* enzyme is Rel.

Atkinson et al. 2011

(<https://www.ncbi.nlm.nih.gov/pmc/articles/PMC3153485/>)

Haurlyuk et al 2015 (<https://www.ncbi.nlm.nih.gov/pmc/articles/PMC4659695/>)

Tanel Tenson

Reviewer #2 (Remarks to the Author):

Hernandez-Valdes et al. studied the regulation of two methionine transporters in the methionine auxotrophic lactic acid bacterium *Lactococcus lactis*. They found that the high-affinity Met-transporter is heterogeneously expressed at low Met concentrations leading to two subpopulations that – surprisingly – are stable over multiple generations. Several layers of regulation modulate expression of the transporter and a T-box riboswitch is accountable for heterogeneity. To the best of my knowledge, this is the first report on a riboswitch involved in heterogeneous gene expression.

The authors use state-of-the-art technology and their manuscript is well written and easy to follow. A limitation of the study is the absence of any mechanistic insights into how the riboswitch gives rise to the remarkable multi-generational phenotypic heterogeneity (see title).

Comments and questions:

1. The evidence that the riboswitch (and not the transcription factor CmhR) is responsible for heterogeneity derives from a mutant that lacks the entire riboswitch. I could not find the exact information how much sequence was deleted but it will certainly be a long stretch and its loss might influence various processes, like transcription elongation, RNA polymerase pausing or transcript stability. Ideally, one would like to see results of strains with less invasive mutations, in which the riboswitch is present but mutated in key residues responsible for its regulatory function.

2. What is the evidence that the authors really look at phenotypic heterogeneity rather than genetic heterogeneity (in other words, point mutations) when the phenotype is stably inherited? It would be comforting to know that the strains indeed are isogenic, as the authors claim. Given that the riboswitch seems to be responsible for heterogeneity, it might be sufficient to sequence across this region in different subpopulations.

3. It is not immediately evident why a part of the population should use the low-affinity transporter when the amino acid is scarce. Apparently, it would be beneficial to have the high-affinity transporter available for acquisition of the limiting nutrient. Here, the stable commitment to an inefficient transporter is most surprising. Instead, one would expect a rapid switch back to the other phenotype. Please comment.

4. In the discussion, the authors provide a reasonable explanation to the question why heterogeneous expression of the Met-transporter might be beneficial. Bet-hedging and division of labor are well-accepted concepts in the field. What is lacking, however, is a reasonable answer to the question how the riboswitch determines heterogeneity. How can it lock expression in a certain state for multiple generations?

5. Figure 6 nicely recapitulates the model how both methionine transporters are regulated. Neither the figure nor the legend refer to heterogeneity, let alone explain it.

6. Is expression of the Met-transporter in the *codY*, *ccpA* and *relA* mutants at low methionine concentrations heterogeneous or homogeneous?

7. Several results shown in the supplementary figures do not appear in the main body of the manuscript, e.g. the *ccpA* mutant in Fig. S10. Please comment on the effect of the regulatory element RE in Fig. S11.

8. Is there a reason why sometimes 1 mM and other times 10 mM were used as high methionine concentration?

Title: Riboswitch gives rise to multi-generational phenotypic heterogeneity in an auxotrophic bacterium

Reviewers' comments

Reviewer #1 (Remarks to the Author):

1. The authors describe phenotypic heterogeneity in bacterial population that has very low switching rate and therefore the phenotypes are maintained over several generations. Although cases of phenotypic heterogeneity have been described previously, the rather stable phenotypes described in the manuscript are novel.

Therefore the paper would be of interest for researchers of the specific field and also for wider audiences.

We thank the reviewer for these supportive comments.

2. The manuscript is very well written. As the Results part is easy to follow I found the beginning of Discussion (lines 288-316) that describes the findings in the manuscript too long and repetitive. The authors can consider shortening this section.

Thanks for pointing this out. We have now streamlined the discussion in several parts, to avoid unnecessary repetition of our results. Instead, we have now added a paragraph to the discussion that describes the dynamics by which both subpopulations originate.

3. An important detail concerns the RelA protein. The RelA was originally described in *Escherichia coli* and there it is not a bifunctional enzyme. Unfortunately some of the diverse bifunctional homologues of RelA that are present in diverse range of bacteria are sometimes also named RelA. This creates considerable confusion. The *Lactococcus* enzyme is Rel.

Atkinson et al. 2011 (<https://www.ncbi.nlm.nih.gov/pmc/articles/PMC3153485/>)

Hauryliuk et al 2015 (<https://www.ncbi.nlm.nih.gov/pmc/articles/PMC4659695/>)

We thank to the reviewer for pointing out this mistake and providing associated references as clarification. We now also noticed that previous studies on “RelA” in *L. lactis* contain the same mistake^{1,2}. To correct for this mistake, we have replaced all cases where we mention “RelA” in the text by “Rel”.

¹ Kok, J. *et al.* The Evolution of gene regulation research in *Lactococcus lactis*. *FEMS microbiology reviews* (2017). doi:10.1093/femsre/fox028

² Rallu, F., Gruss, A., Ehrlich, S. D. & Maguin, E. Acid- and multistress-resistant mutants of *Lactococcus lactis*: Identification of intracellular stress signals. *Mol.*

Reviewer #2 (Remarks to the Author):

Hernandez-Valdes *et al.* studied the regulation of two methionine transporters in the methionine auxotrophic lactic acid bacterium *Lactococcus lactis*. They found that the high-affinity Met-transporter is heterogeneous expressed at low Met concentrations leading to two subpopulations that – surprisingly – are stable over multiple generations. Several layers of regulation modulate expression of the transporter and a T-box riboswitch is accountable for heterogeneity. To the best of my knowledge, this is the first report on a riboswitch involved in heterogeneous gene expression. The authors use state-of-the-art technology and their manuscript is well written and easy to follow. A limitation of the study is the absence of any mechanistic insights into how the riboswitch gives rise to the remarkable multi-generational phenotypic heterogeneity (see title).

We thank the reviewer for this supportive feedback. We are also grateful for the constructive criticisms, which we believe has enormously helped to improve our manuscript. As suggested by the reviewer, we have now examined the heterogeneity in *met* expression in mutants of global regulators as well as targeted mutations in the riboswitch. Altogether, these mutants clarified how the phenotypic heterogeneity comes about. The details of which, we outline below.

Comments and questions:

1. The evidence that the riboswitch (and not the transcription factor CmhR) is responsible for heterogeneity derives from a mutant that lacks the entire riboswitch. I could not find the exact information how much sequence was deleted but it will certainly be a long stretch and its loss might influence various processes, like transcription elongation, RNA polymerase pausing or transcript stability. Ideally, one would like to see results of strains with less invasive mutations, in which the riboswitch is present but mutated in key residues responsible for its regulatory function.

We now highlight in Fig. S17a what part of the DNA sequence was deleted in our mutant that lacks the entire regulatory element. We also agree with the reviewer that it would be interesting to examine less invasive mutations in the riboswitch. Therefore, we now examined four highly-targeted mutations in conserved residues of the T-box riboswitch. The mutations specifically target (i) the stability of the stem I domain, (ii) the formation of the anti-termination complex, by either affecting tRNA binding to the T-box or the stability of the anti-terminator, and (iii) the formation of the terminator hairpin. Fig. 6 and Fig. S17b visualize the different mutations and show the resulting *met* expression using both microscopy and flow cytometry. Overall, the results strongly corroborate our previous findings that the riboswitch gives rise to the phenotypic heterogeneity. None of the mutations show phenotypic heterogeneity and two mutations show particularly interesting results. Namely, the mutant preventing tRNA binding to the riboswitch shows expression patterns that match exactly with the GFP⁻ cells in the wild type, whereas the mutant that prevents the formation of the terminator hairpin shows expression patterns that match the GFP⁺ subpopulations in the wild type. These mutants thereby confirm that the GFP⁻ subpopulation originates from the successful formation of the terminator hairpin, that lowers *met* expression through transcriptional attenuation, whereas the GFP⁺ subpopulation originates from the formation of the anti-terminator complex by binding of uncharged tRNAs to the T-box element. Further information regarding the strains is given in Supplementary Table 2.

2. What is the evidence that the authors really look at phenotypic heterogeneity rather than genetic heterogeneity (in other words, point mutations) when the phenotype is stably inherited? It would be comforting to know that the strains indeed are isogenic, as the authors claim. Given that the riboswitch seems to be responsible for heterogeneity, it might be sufficient to sequence across this region in different subpopulations.

We have now sequenced the 5' UTR end of the *met* operon by PCR in bacterial colonies of each phenotype: two GFP+ colonies, two GFP- colonies and the GFP+ and GFP- sectors of a switching colony. No mutations in the promoter regions of the *met* operon were detected. To also completely rule out mutations elsewhere in the genome, we have also performed whole-genome sequencing in one GFP+ colony, one GFP- colony and the GFP+ and GFP- sectors of a switching colony. Also in these samples, mutations were absent, which confirms our expectation that the GFP- and GFP+ subpopulations are isogenic. All results are summarized in Supplementary Fig. 5. We will upload all sequencing data upon publication.

3. It is not immediately evident why a part of the population should use the low-affinity transporter when the amino acid is scarce. Apparently, it would be beneficial to have the high-affinity transporter available for acquisition of the limiting nutrient. Here, the stable commitment to an inefficient transporter is most surprising. Instead, one would expect a rapid switch back to the other phenotype. Please comment.

Good point. We first want to emphasize that the GFP- cells do still express the *met* operon, although on a much lower level than the GFP+ cells (Fig. 2f, 5d). In fact, we know that the GFP- cells partly rely on the expression of the *met* operon, since a knockout of the *met* operon does not support growth at the lowest methionine concentrations, which indicates that cells cannot fully compensate for the absence of the *met* operon by increasing the expression of the low affinity transporter (Fig. S7, S8). We realized that the schematic summary figure was a bit confusing in this regard and have therefore considerably revised this figure to highlight that both the GFP- and GFP+ cells express the *met* operon, but that their expression levels differ substantially (see new Fig. 7).

Despite the different expression levels of the *met* operon, the fitness differences between the GFP+ and GFP- colonies are surprisingly small. We therefore hypothesized that the GFP- cells compensate for the lack of *met* expression by increasing the expression of the low-affinity transporter (BcaP). Indeed, when we knockout CodY (a repressor of *bcaP*), we lose the subpopulation of cells that highly express the *met* operon (i.e. GFP+ cells), which suggests that the low-affinity transporter can partly (not entirely) compensate for the high-affinity transporter. From this perspective, one does not necessarily expect a high switching rate, because two alternative routes for methionine uptake only show minimal differences.

To clarify our arguments, we have now substantially changed the text (page 8, lines 197-205), as well as adjusted Fig. 7.

4. In the discussion, the authors provide a reasonable explanation to the question why heterogeneous expression of the Met-transporter might be beneficial. Bet-hedging and division of labor are well-accepted concepts in the field. What is lacking, however, is a reasonable answer to the question how the riboswitch determines heterogeneity. How can it lock expression in a certain state for multiple generations?

We thank the reviewer for emphasizing this shortcoming. Both in reply to this comment and comment 6, we have evaluated *met* expression in different knockout mutants of the global regulators. As it has been shown before that heterogeneity in *L. lactis* can result from cells that are locked in distinct physiological states³, we hypothesized that the differential *met* expression in the GFP- and GFP+ populations could reflect such physiological states as well. Strikingly, the knockouts of all global regulators (*codY*, *rel*, *ccpA*) abolishes the bimodal gene expression of the *met* operon as seen in the wild type (for details see comment 6 below; Fig. S11). The *rel* knockout is particularly intriguing, because it shows expression distributions of the *met* operon across the different methionine concentrations that correspond exactly to those of the GFP- subpopulation in the wild type. The *rel* knockout furthermore gave the same expression patterns as the riboswitch mutant that prevents binding of uncharged tRNAs. Given that Rel is activated by uncharged tRNAs⁴, these results suggest that the stringent response is activated in cells with high *met* expression, which could lock them in this state. A double knockout mutant that we made previously, indeed shows that the *rel* knockout exerts its effect on *met* expression via the riboswitch (Fig. S12), as opposed to, for example, the expression or activity of the transcription factors (CmhR).

Altogether, we therefore envision the following scenario. Upon methionine depletion cells need to increase their methionine uptake rates. In some cells, CodY repression of the *bcaP* operon is released timely, due to which the low-affinity transporter can sequester enough methionine from the environment to support growth (these cells will also increase the expression of the *met* operon, due to activity of the transcription factor). Consequently, these cells will not experience methionine starvation that leads to uncharged tRNAs (Note that in the absence of the low-affinity transporter all cells will highly express the *met* operon; see Fig. 4, S18). In contrast, other cells might not be able to increase uptake rates timely and will experience methionine starvation leading to uncharged tRNAs. These tRNAs trigger both high expression of the *met* operon (via the riboswitch) and the stringent response. We think that the stringent response could be responsible for keeping cells locked in a distinct physiological state (note that in *L. lactis* the stringent response does not repress CodY activity by changing the GTP levels, in contrast to *Bacilli* species^{5,6}). This is supported by the fact that GFP+ cells readily lower *met* expression when exposed to nutrient rich conditions (which suppress the stringent response).

We now extensively adjusted both the results and discussion sections to incorporate the new results.

³ Solopova, A. *et al.* Bet-hedging during bacterial diauxic shift. *Proc. Natl. Acad. Sci.* (2014). doi:10.1073/pnas.1320063111

⁴ Chang, D. E., Smalley, D. J. & Conway, T. Gene expression profiling of Escherichia coli growth transitions: An expanded stringent response model. *Mol. Microbiol.* (2002). doi:10.1046/j.1365-2958.2002.03001.x

⁵ Petranovic, D. *et al.* Intracellular effectors regulating the activity of the Lactococcus lactis CodY pleiotropic transcription regulator. *Mol. Microbiol.* (2004). doi:10.1111/j.1365-2958.2004.04136.x

⁶ Geiger, T. & Wolz, C. Intersection of the stringent response and the CodY regulon in low GC Gram-positive bacteria. *International Journal of Medical Microbiology* (2014). doi:10.1016/j.ijmm.2013.11.013

5. Figure 6 nicely recapitulates the model how both methionine transporters are regulated. Neither the figure nor the legend refer to heterogeneity, let alone explain it.

We now strongly adjusted this figure in line with the reasoning above (see answer to comment 4). We also adjusted the corresponding figure caption and the discussion of the figure in the discussion section. We hope that the figure is now sufficiently clear, but we are of course happy to make further adjustments when the reviewer has additional ideas on how to improve it.

6. Is expression of the Met-transporter in the *codY*, *ccpA* and *relA* mutants at low methionine concentrations heterogeneous or homogeneous?

We thank the reviewer for this valuable comment, which turned out to be critical for our evaluation for the different subpopulations of GFP- and GFP+ cells (see comment 4). We have now examined the heterogeneity in each of these mutants using flow cytometry, and – as pointed out above – this revealed that none of the mutants show a bimodal expression peak (see Supplementary Fig. 11, note that the expression values are shown on a log-scale). In addition to the role of Rel, which we discussed above (comment 4), we hypothesize that the *codY* mutant prevent heterogeneity, because CodY normally represses the expression of *bcaP*. Thus, in the absence of CodY all cells will highly express the low-affinity transporter, which prevents methionine starvation and hence the rise of uncharged Met-tRNAs at low methionine concentrations.

7. Several results shown in the supplementary figures do not appear in the main body of the manuscript, e.g. the *ccpA* mutant in Fig. S10. Please comment on the effect of the regulatory element RE in Fig. S11.

We thank the reviewer for pointing out these mistakes. We now briefly mention the *ccpA* knockout mutant in the text (page 10, lines 233-239). We tested this mutant, because of its importance to sulfur metabolism, which indirectly links to amino acid uptake⁷.

Regarding the regulatory element RE in Fig. S11 (currently Fig. S12), we observed in our original manuscript that the *rel* knockout mutant most strongly reduced *met* expression at low methionine concentration. We were therefore curious to see if Rel exerts an effect through the riboswitch (as opposed to for example the activity of the transcription factor), which is the case. This was particularly interesting, because it is known for a while that Rel can promote amino acid uptake, but the mechanisms by which this happens is not completely understood for *L. lactis*^{8,9}. Our new results suggest that the stringent response indirectly promotes methionine uptake by keeping cells in the high *met* expression state (i.e. in the *rel* knockout mutant, only GFP- cells are observed). It is an exciting future challenge to determine how exactly this effect comes about. We have now integrated the results of Fig. S11 (current Fig. S12) in the final part of the results section.

⁷ Gaudu, P., Lamberet, G., Poncet, S. & Gruss, A. CcpA regulation of aerobic and respiration growth in *Lactococcus lactis*. *Mol. Microbiol.* (2003). doi:10.1046/j.1365-2958.2003.03700.x

⁸ Gaca, A. O., Abranches, J., Kajfasz, J. K. & Lemos, J. A. Global transcriptional analysis of the stringent response in *Enterococcus faecalis*. *Microbiol. (United Kingdom)* (2012). doi:10.1099/mic.0.060236-0

⁹ Kok, J. *et al.* The Evolution of gene regulation research in *Lactococcus lactis*. *FEMS microbiology reviews* (2017). doi:10.1093/femsre/fux028

8. Is there a reason why sometimes 1 mM and other times 10 mM were used as high methionine concentration?

Good point. We initially examined *L. lactis* cells using time-lapse microscopy, where a methionine concentration of 1 mM was enough to suppress GFP expression (in our reporter strain). It was only when we also performed flow cytometry that we noticed that a methionine concentration of 10 mM gave even slightly lower expression values than a concentration of 1 mM (notice that these are only small differences, since they occur at the lower end of the log-scale). For this reason, we have occasionally used different concentrations in our experiments. Yet, we would like to point out that this has absolutely no consequences for the results of our manuscript. We now adjusted the text to highlight what methionine concentration we used every time we mention “high methionine concentration”.

REVIEWERS' COMMENTS:

Reviewer #2 (Remarks to the Author):

The revised version presents new experiments and the authors responded adequately to all comments.